# Rice Husk Biochars Modified with Magnetized Iron Oxides and Nano Zero Valent Iron for Decolorization of Dyeing Wastewater

**Bao-Son Trinh** [1,*]**, Phung T. K. Le** [2]**, David Werner** [3]**, Nguyen H. Phuong** [2,4] **and Tran Le Luu** [5]

[1]  Institute for Environment and Resources, Vietnam National University Ho Chi Minh City,
    Ho Chi Minh City 75308, Viet Nam

[2]  Ho Chi Minh City University of Technology, Vietnam National University Ho Chi Minh City,
    Ho Chi Minh City 72506, Viet Nam; phungle@hcmut.edu.vn

[3]  School of Engineering, Newcastle University, Newcastle upon Tyne, NE1 7RU, UK;
    david.werner@newcastle.ac.uk

[4]  Tay Nguyen University, Buon Ma Thuot City 63000, Viet Nam; phuongmt4@gmail.com

[5]  Department of Mechatronics and Sensor Systems Technology, Vietnamese-German University;
    Binh Duong 75000, Viet Nam; luu.tl@vgu.edu.vn

*   Correspondence: bao-son.trinh@hcmier.edu.vn; Tel.: +84-28-2253-8586

**Abstract:** This study investigated if biochar, a low-cost carbon-rich material, can be modified with reactive materials for decolorization of dyeing wastewater. Two types of rice husk biochars were produced by using different processes of gasification and pyrolysis in limited air condition. The biochars were first magnetized and then modified with nano-scale zero-valent iron (nZVI) to achieve the final products of magnetic-nZVI biochars. Batch experiments were conducted to investigate the efficiency of the modified biochars for reducing color of the reactive dyes yellow (RY145), red (RR195), and blue (RB19) from dyeing solutions. Results showed that color removal efficiency of the modified biochars was significantly enhanced, achieving the values of 100% for RY145 and RR195 and ≥65% for RB19, while the effectiveness of the original biochar was significantly lower. In addition, with increasing dose of the modified biochars, the color removal efficiency increased accordingly. In contrast, when the dose of nZVI was increased beyond a certain value then its color removal efficiency decreased accordingly. It is reported that the magnetic-nZVI rice husk biochars effectively removed the reactive dyes. The impregnation of nZVI particles on the biochar surface spatially separates the nZVI particles, prevents its aggregation and therefore enhances the decolorization efficiency.

**Keywords:** biochar; full-scale gasification; pyrolysis; rice husk; nano zero-valent iron

---

## 1. Introduction

Biochar is an organic product formed through the heating of biomass to above 250 °C without oxygen, known as pyrolysis process, or with limited air with the stoichiometric air-biomass ratio of 0.15–0.28, known as gasification process [1]. As a carbon-rich material, the morphology of biochar is characterized by a larger pore space than its original biomass due to the dehydration process at low temperature of 100–250 °C and the loss of cellulose, lignin and hemicellulose at higher temperature of 250–700 °C [2]. This porous structure of biochar creates adsorption sites to physically retain the pollutants [3–5]. Biochars can be functionalized by chemically combining them with other functional elements e.g., magnetized iron oxides ($Fe_3O_4$), maghemite ($Fe_2O_3$, $\gamma$-$Fe_2O_3$) or nano zero valent iron (nZVI), for enhancement of its original properties by adding features such as magnetism and/or

reactivity [6–8]. Application of magnetized carbon-based materials as adsorbents is of interest in industrial wastewater treatment processes because they can be easily separated with an external magnet from the treated water flow [9,10]. Application of nZVI particles to remove various contaminants (e.g., chlorinated, halogenated aliphatic, nitrates, nitro aromatic carbons, phenols, heavy metals, inorganic species, explosives, and pesticides) has been reviewed extensively [11,12]. In addition, application of carbon-based materials modified with nZVI is increasingly being considered for organic pollution remediation not only in the groundwater environment but also in textile dyeing wastewater treatment [13,14]. A combination of nZVI and carbon-based materials is expected to enhance the advantages of both. However, the properties and performance of such combination materials needs to be further studied. While the application of various biochars derived from pyrolysis processes has been widely reported [10,15], the application of biochar derived from a full-scale gasification process has not yet been sufficiently considered.

Numerous biomass sources could be used as input for biochar production, e.g., corn cobs, coconut shell, ground coffee powder, coffee hull, or even livestock manure. However, rice husk has emerged as an abundant biomass source in Vietnam which could be utilized to produce biochar. Rice husk possesses a natural polymer as it contains approximately 40% cellulose, 30% lignin group, and the remaining 20% consist of silica, adsorbed water, alkaline minerals, and other trace elements [15,16]. In 2017, Vietnam produced 43 million tons of paddy rice and became the fifth largest rice producing country on Earth [17]. The Mekong Delta in the South of Vietnam is the largest paddy field of the country which has produced more than 56% of the total amount of paddy rice production [18]. According to Fernandes, Calheiro [19] the proportion of rice husk to bulk grain weight is approximately 20% and, as a result, the annual amounts of rice husk were estimated to be 9 million tons for the whole country and 5 million tons for the Mekong Delta only. Rice husk was therefore used as a biomass resource to produce biochar in this study.

The textile dyeing industry has been quickly developing in Vietnam. Even though considerably contributing to the national economy, it is also causing significant environmental impacts, especially to the aquatic environment, due to the large amounts of heavy polluted wastewater being produced. Effluents from textile dyeing wastewater are complex and often consist of dyes, alkalis, organic and inorganic salts, acids, and heavy metals [20]. In an attempt to prevent color pollution from dyeing wastewater, various methods have been proposed, for instance, biochars made by bamboo [21] and other agricultural wastes [22] were used for adsorption of acid dyes (Acid-Blue-25 and Acid-Yellow-17) and methylene blue dye. Resin supported nZVI was used for decoloration of Acid-Blue-113 azo-dye [23]. Degradation of textile dyes using nZVI supported by various materials such as resin, nickel, zinc, bentonite, biopolymer, kaolin, rectorite, nickel-montmorillonite, bamboo, cellulose, biochar, graphene, and clinoptilolite was also reviewed [13]. However, application of magnetic-nZVI rice husk biochar derived from a full-scale gasifier for removal of color from dyeing wastewater has not yet been investigated.

This study aimed to produce rice husk biochars derived from a full-scale gasification and a lab-scale pyrolysis process. The biochars were then modified with magnetized iron oxides to create an intermediate product of magnetic biochars. Zero valent iron was synthesized on these intermediate materials to achieve the final product of magnetic-nZVI biochars. Color removal efficiency of the magnetic-nZVI biochars and their original materials of biochar and nZVI for the representative reactive dyes was assessed by designing and conducting batch experiments.

## 2. Materials and Methods

Rice husk: Rice husk was collected from the Ecofarm Co. Ltd. in Long An, a Mekong Delta province in the South of Vietnam. Rice husk was dried at room temperature (25–30 °C) for at least 48 h before use.

Biochars: Two types of rice husk biochars were produced by using full-scale gasification and lab-scale pyrolysis processes as described below.

- Full-scale gasification rice husk biochar: A full-scale gasifier (BiGchar 2200 fast rotary hearth, φ = 2.2 m, h = 2 m, nominal capacity of 300 kg biochar/h, designed and fabricated by Pyrocal Pty Ltd, Wellcamp, Queensland, Australia) was used to produce gasification biochar (Figure 1). The gasifier has four chambers equipped temperature and air controllers. Briefly, rice husk was continuously fed on top and moved down by a rotary hand in every chamber. The top three chambers were heated in a range of temperature of 300–650 °C by the energy of the gasification process under air controlled conditions. The bottom chamber was used for cooling. Retention time of rice husk in the gasifier was approximately 1 h. Gasification biochar after cooling was collected by a screw conveyor system and labeled as BCgas.
- Lab-scale pyrolysis rice husk biochar: A lab-scale closed furnace (EF 11/8B, Lenton, Hope Valley, UK) and other associated equipments including the closed-steel cylinder and its components (Hoang Ha Co. Ltd, Binh Duong, Vietnam) were used for producing pyrolysis biochar (Figure 2). Briefly, rice husk (40 g) was transferred to the steel-cylinder (φ = 2.5 cm, L = 10 cm, Figure 2b,c), tightly closed at one end and screwed tight by hand at the other end with a piston-screw mechanic system. This step aimed to minimize available air in the cylinder. It was then placed in the furnace and heated (heating rate of approx. 40 °C/min) to a pre-set temperature and retain for 1 h. Two thresholds of temperature were set to be 400 °C and 800 °C. The furnace was finally turned off for cooling down for approximately 4 h. Biochars at 400 °C and 800 °C were collected and labeled as BC400 and BC800, respectively.

The yield of biochar was calculated as follows:

$$Biochar\ yield\ (\%) = \frac{mass\ of\ biochar\ (g)}{dried\ mass\ of\ biomass\ (g)} \times 100 \qquad (1)$$

Ash content of biochar was determined by a dry combustion method [24]. In brief, about 5.0 g of biochar was heated at 500 °C for 8 h. The crucible was then cooled to room temperature and reweighed. The ash content was calculated as follows:

$$Ash\ content\ (\%) = \frac{weight\ of\ ash\ (g)}{dried\ mass\ of\ biochar\ (g)} \times 100 \qquad (2)$$

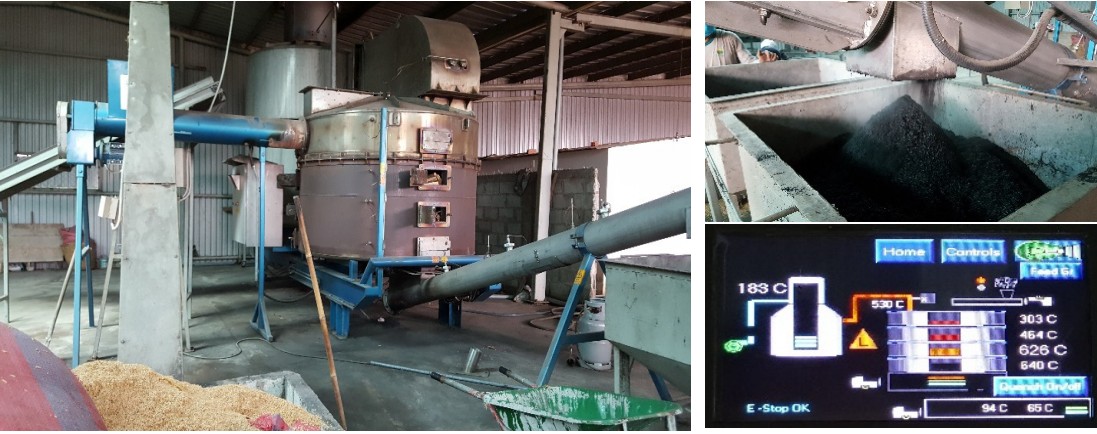

**Figure 1.** Full-scale gasification system (BiGchar 2200 fast rotary hearth, Pyrocal Pty Ltd, Wellcamp, Queensland, Australia) for biochar production. Owned by Ecofarm Co. Ltd., Long An province, Vietnam.

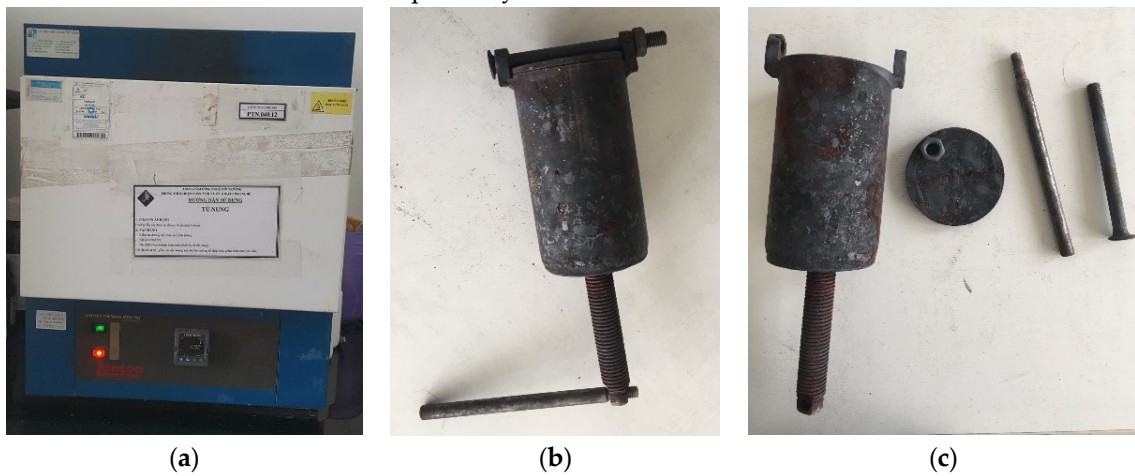

**Figure 2.** (**a**) Lab-scale closed furnace (EF 11/8B, Lenton, Hope Valley, UK) for biochar production; (**b**) closed-steel cylinder for containing biomass; (**c**) components of the closed-steel cylinder.

pH of biochar was determined according to Savova, Apak [25]. About 4.0 g biochar was mixed with 100 mL deionized water in a conical flask. The flask was boiled for 5 min and left for cooling to room temperature. The supernatant was decanted and its pH value was determined by using a pH meter (ProDSS, Xylem Inc, Rye Brook, NY, USA).

Zero point of charge pH ($pH_{ZPC}$) of biochar was determined following previous work [26]. To a series of Erlenmeyer flasks (100 mL), 40 mL of $NaNO_3$ (0.1 M, Scharlau, Sentmenat, Barcelona, Spain) solution was added and the pH was adjusted by using $HNO_3$ (Scharlau, Sentmenat, Barcelona, Spain) and NaOH (Scharlau, Sentmenat, Barcelona, Spain) with different concentrations (0.1–1 M) to give initial pH ($pH_i$) of 2, 4, 6, 8, and 10. To each flask, 0.1 g biochar was added and the suspensions were agitated at a stirring speed of 250 rpm overnight at room temperature. In the next day, final pH ($pH_f$) of the suspensions was recorded and the difference between the final and the initial pH ($\Delta pH$) was plotted against the $pH_i$. The pH value where net surface charge was zero or $\Delta pH = 0$, is considered to be the $pH_{ZPC}$ of the materials.

Magnetic biochars: The original biochars, including BCgas, BC400, and BC800, were magnetized by a wet precipitation method [6,9,27,28] with a few modifications. Briefly, $Fe^{2+}/Fe^{3+}$ solution was prepared by dissolving $FeSO_4 \cdot 7H_2O$ (3.66 g, Scharlau, Sentmenat, Barcelona, Spain) and $FeCl_3 \cdot 6H_2O$ (6.66 g, Scharlau, Sentmenat, Barcelona, Spain) in deionized water (200 mL). The biochars (5 g for each) were individually transferred to the $Fe^{2+}/Fe^{3+}$ solution. The mixtures were heated to 65 °C and retained for 3 h in agitated condition. The mixtures were then cooled down to approximately 40 °C and pH was raised to 10–11 by adding NaOH 5 M. Dark brown particles were observed at this alkaline pH value. The mixtures were agitated for an additional 1 h and left to settle overnight. Supernatants were decanted and the precipitated solid was washed with deionized water and rinsed with ethanol for 3 times. The magnetized solids were dried at 80 °C overnight. The dried solids were mixed again in deionized water and a magnet was used to separate and collect only the magnetized biochar particles. The remaining particles which were not attached to the magnet were discarded along with the solution. The presence of magnetic minerals in the final products was confirmed by the use of a magnet. The intermediate magnetized products were collected and labelled as BCgas-m, BC400-m, and BC800-m, respectively.

Nano zero valent iron (nZVI): Lower surface area nZVI was synthesized by using traditional reduction methods [29,30]. Briefly, a strong reductant of sodium borohydride ($NaBH_4$, Scharlau, Sentmenat, Barcelona, Spain) was used to reduce ferrous ion ($Fe^{2+}$) to zero valent iron ($Fe^0$) as described by Equation (3). At first, $FeSO_4 \cdot 7H_2O$ (20 g) was dissolved with deionized water (100 mL) in a 500 mL beaker. The reductant $NaBH_4$ (2 g) was also dissolved with deionized water (100 mL) in a 250 mL beaker. The $NaBH_4$ solution was transferred to a burrette for reaction with the $Fe^{2+}$ solution by

dropping $NaBH_4$ (approx. 30 drops/min) to the $Fe^{2+}$ solution. The mixture was agitated by a magnetic stirrer for 1 h at room temperature (28 °C). The supernatant was decanted and the precipitated solid was washed with deionized water and rinsed with acetone for 3 times. The solid was dried at 105 °C for 24 h under a stream of $N_2$ at 500 mL/min. The nanoscale zero valent iron product was finally collected and labeled as nZVI.

$$Fe^{2+} + 2BH_4^- + 6H_2O \rightarrow Fe^0 \downarrow + 2B(OH)_3 + 7H_2 \uparrow \tag{3}$$

Magnetic-nZVI biochars: The magnetic biochars of BCgas-m, BC400-m and BC800-m were modified with nZVI by using a reduction method [8] which is similar to the above method for synthesizing nZVI. Briefly, the magnetic biochars (5 g) were individually transferred to a $FeSO_4 \cdot 7H_2O$ solution (20 g in 100 mL). The mixtures were sonicated for 2 h and vigorously shaken for 48 h at 150 rpm on an orbital shaker (SA300, Yamato Scientific America Inc., Santa Clara, CA, USA) to ensure $FeSO_4$ molecules were equally distributed in the pore-space structure of the materials. A strong reductant of sodium borohydride ($NaBH_4$) was used to reduce ferrous ion ($Fe^{2+}$) to zero valent iron ($Fe^0$) as described by Equation (3). The mixtures were reacted with $NaBH_4$ (2 g in 100 mL) drop-wise (30 drops/min). The reacted solutions were stirred vigorously at room temperature (28 °C) for 1 h. The supernatants were decanted and the precipitated solids were washed with deionized water and rinsed with acetone (3 times). The solids were dried at 105 °C for 24 h under a stream of $N_2$ at 500 mL/min. The final products were collected and labeled as BCgas-m-nZVI, BC400-m-nZVI and BC800-m-nZVI.

Morphology of the materials was inspected by using the JEM-1400 Transmission Electron Microscope (JOEL USA Inc., Peabody, MA, USA). Specific surface area of the materials was determined using the Brunauer–Emmett–Teller technique with Autosorb-1 surface area analyzer (Quantachrome Instruments, Boynton Beach, Florida, USA).

Chemicals: The dyes of C.I. Reactive Red 195, labeled as RR195 (CAS 93050-79-4, commercial grade ≥ 95.0%), C.I. Reactive Yellow 145, labeled as RY145 (CAS 93050-80-7, commercial grade ≥ 95.0%), and C.I. Reactive Bule 19, labeled as RB19 (CAS 2580-78-1, commercial grade ≥ 95.0%) were purchased from Tan Duy Phat Co. Ltd. in Ho Chi Minh city, Vietnam. The dyes were chosen based on their common use in textile dyeing industry [13,20,31]. The dyes RR195 and RY145 are classified as single azo colors with heavier molecular weights (Table 1), while the dye RB19 is classified as anthraquio color with lighter molecular weight (Table 1). Their empirical formulae are presented in Table 1. Sodium borohydride ($NaBH_4$, CAS 16940-66-2, analytical grade 98%), iron (II) sulfate heptahydrate ($FeSO_4.7H_2O$, CAS 7782-63-0, analytical grade 99.55%), iron (III) chloride hexahydrate ($FeCl_3.6H_2O$, CAS 10025-77-1, analytical grade 99.9%), acetone ($C_3H_6O$, CAS 76-74-1, analytical grade 99.5%), and ethanol ($C_2H_5OH$, CAS 64-17-5, analytical grade 99.9%) were purchased from Scharlau chemical. Advantec filter sheet-type 1 (cellulose, 6 μm) and syringe filters (polyethersulfone membrane, 0.45 μm) were purchased from Toyo Roshi Kaisha, Ltd., Tokyo, Japan.

Dyeing stock solutions: For investigation of color removal efficiency of the materials, the dyeing stock solutions RY145, RR195, and RB19 were prepared in order to achieve the final color thresholds of approx. 400 Pt-Co. Different amounts of RY145 (0.01152 g), RR195 (0.02784 g), and RB19 (0.10410 g) were dissolved in deionized water (1000 mL), sonicated in an ultra-sonic bath for 1 h at 28 °C (Elmasonic S 100 H, 37 kHz), and shaken for 2 h at 150 rpm (orbital shaker SA300, Yamato Scientific America Inc., Santa Clara, CA, USA) to give the absolute concentrations of $11.2 \times 10^{-6}$ M, $24.5 \times 10^{-6}$ M, and $166.1 \times 10^{-6}$ M, respectively. Color thresholds of the final stocks RY145, RR195, and RB19 were of 404.5, 408.8, and 410.9 Pt-Co determined by an UV-1800 spectrophotometer (Shimadzu Scientific Instruments, Columbia, MD, USA) at different wavelengths for maximum absorbance of 419 nm [32], 517 nm [33], and 592 nm [34], respectively. UV-VIS is a reliable method to confirm dye concentration [32–34]. It is noted that the National Technical Regulation on Industrial Wastewater of Vietnam [35] does not provide an absolute concentration threshold (in mg/L or molar unit) for a specific dye but it provides a color threshold (in Pt-Co unit) for industrial wastewater being discharged into receiving waters.

For instances, the maximum values of color threshold of industrial wastewater in Columns A and B are stated as 50 and 150 Pt-Co, respectively. According to this Regulation, the experiments were designed for investigation of the color removal efficiency of the modified biochar materials based on the color threshold (in Pt-Co unit) of a solution rather than the absolute concentrations (in mg/L or molar unit).

**Table 1.** Empirical and molecular formulae of the reactive dyes.

| Chemicals | CAS | Empirical Formula | Weight (g/mol) | Molecular Formula |
|---|---|---|---|---|
| C.I. Reactive Red 195 (RR195) | 93050-79-4 | $C_{31}H_{19}ClN_7Na_5O_{19}S_6$ | 1136.3 |  |
| C.I. Reactive Yellow 145 (RY145) | 93050-80-7 | $C_{28}H_{20}ClN_9Na_4O_{16}S_5$ | 1026.25 |  |
| C.I. Reactive Blue 19 (RB19) | 2580-78-1 | $C_{22}H_{16}N_2Na_2O_{11}S_3$ | 626.55 |  |

Assessing the loss of color due to the use of a centrifuge tube (PE, 50 mL) and syringe filter (PES, 0.45 µm): For the centrifuge tube, a volume (25 mL) of every dyeing stock solutions RY145, RR195, and RB19 with initial color thresholds of 404.5, 408.8, and 410.9 Pt-Co was transferred to the sterile tubes and shaken for 0 (control treatment) and 1 h at 150 rpm (orbital shaker SA300, Yamato Scientific America Inc., Santa Clara, CA, USA). The tubes were then centrifuged for 30 min at 4000 rpm (Rotofix 32A, Andreas Hettich GmbH & Co. KG, Tuttlingen, Germany) and the color threshold (in Pt-Co) was determined by an UV-1800 spectrophotometer (Shimadzu Scientific Instruments, Columbia, MD, USA) at different wavelengths as stated in "Dyeing stock solutions" section. After a period of 1 h contacting time, color of the RY145, RR195, and RB19 stocks decreased from 404.5 ± 0 to 389.0 ± 6.90 Pt-Co, from 408.8 ± 0 to 403.8 ± 1.2 Pt-Co, and from 410.9 ± 0 to 405.8 ± 2.0 Pt-Co, equivalent to the losses of 3.8 ± 1.7%, 1.22 ± 0.28%, and 1.2 ± 0.5% (n = 3), respectively. For the syringe filters, color thresholds of the RY145, RR195, and RB19 stocks were determined before and after filtration, with the decreases from 404.5 ± 0.0 to 398.8 ± 1.2 Pt-Co, 408.8 ± 0.0 to 402.9 ± 1.39 Pt-Co, and from 410.9 ± 0.0 to 404.9 ± 1.15 Pt-Co, equivalent to the losses of 1.40 ± 0.30%, 1.42 ± 0.34%, and 1.46 ± 0.28% (n = 3), respectively. These results showed that the color loss of the RY145, RR195, and RB19 stocks due to the use of a syringe filter and centrifuge tube were not significant (<5%).

*Experimental Methods for Investigation of Color Removal Efficiency (η,%) of the Modified Biochars*

Batch experiments for removal of the reactive dyes RY145, RR195, and RB19: Known amounts of nZVI, BCgas-m-nZVI, BC400-m-nZVI, and BC800-m-nZVI were separately mixed with the individual stocks RY145, RR195, and RB19. Another known amount of BC800 was also mixed with RR195 stock. A volume of the stocks (25 mL) was transferred to centrifuge tubes (PE, 50 mL). With the materials of nZVI, BCgas-m-nZVI, BC400-m-nZVI, and BC800-m-nZVI, different amounts were added in the tubes to achieve different doses (g) of the materials in a volume (L) of the stocks as follows: 0.0 (control treatment), 0.25, 0.5, 1.0, 1.5, and 2.0 g/L for the treatments with RY145; 0.0, 0.5, 1.0, 1.5, and 2.0 g/L for the treatments with RR195; and 0.0, 1.0, 2.0, 4.0, 6.0, 8.0, 10.0, and 12.0 g/L for the treatments with RB19. All treatments were replicated three times (n = 3). With BC800, different amounts were added in the

tubes to achieve different doses (g) of BC800 in a volume (L) of RR195 stock as 0.0 (control treatment), 0.5, 5.0, and 10.0 g/L for the treatments with RR195. Screening tests were also conducted in advance to determine the above optimum ranges of dose. Then the tubes were tightly closed with caps, placed horizontally in a plastic box and vigorously shaken for 1 h at 150 rpm (orbital shaker SA300, Yamato Scientific America Inc., Santa Clara, CA, USA). The tubes were centrifuged for 30 min at 4000 rpm (Rotofix 32A). Finally, the supernatant from every tube was collected and filtered (polyethersulfone syringe filter, 0.45 μm). The color threshold (in Pt-Co) of every treated solution was measured by by an UV-1800 spectrophotometer (Shimadzu Scientific Instruments, Columbia, MD, USA) at different wavelengths as stated in "Dyeing stock solutions" section.

Color removal efficiency (η %) of the materials was calculated as follows:

$$\eta(\%) = \frac{C_0 - C_1}{C_0} \tag{4}$$

where $C_0$—the initial color threshold of the stocks, (Pt-Co); $C_1$—the color threshold of the stocks after 1 h contacting time, (Pt-Co).

## 3. Results and Discussion

### 3.1. Physico-Chemical Properties of the Materials

Basic physico-chemical properties of the original materials, including BC400, BC800, BCgas, and nZVI, and the modified materials, including BC400-m-nZVI, BC800-m-nZVI, and BCgas-m-nZVI were determined and reported in Table 2. The B.E.T. specific surface area (SSA) of the gasified biochar (BCgas) achieved the highest value of 251.11 m²/g while the SSA of the pyrolyzed biochars of BC400 and BC800 achieved lower values of 141.23 and 213.03 m²/g, respectively. Agitation condition during the heating period could support the charring process [1]. Better agitation condition of the full-scale gasifier (having four chambers equipped with rotary hands, Figure 1) than the stationary condition of the lab-scale furnace (Figure 2) may result in the higher SSA value even with similar retention time of 1 h. The SSA values of this study is slightly lower than the SSA values of rice husk biochars reported by other studies. For instance, the SSA value of 251.11 m²/g of BCgas from this study is lightly lower than the SSA value of 261.72 m²/g of the rice husk biochar published by Claoston, Samsuri [15]. After modified by magnetized iron oxides and nZVI, the SSA of BCgas-m-nZVI, BC400-m-nZVI, and BC800-m-nZVI was significantly lower than the one of the original biochars, achieving the values of 192.07, 132.44, and 181.92 m²/g, respectively. The deposition of micro- and nano-particles of magnetized iron oxides and nZVI on the porous structure of the biochars could result in the decrease of their SSA values. In addition, iron oxides, as part of the mass, were reported to have lower surface area than the one of biochar [36]. Hence the SSA of the modified biochars should also be lower.

**Table 2.** Physico-chemical properties of the materials of full-scale gasification rice husk biochar (BCgas), 400 °C lab-scale pyrolyzed rice husk biochar (BC400), 800 °C lab-scale pyrolyzed rice husk biochar (BC800), nano zero valent iron (nZVI), magnetic-nZVI BCgas (BCgas-m-nZVI), magnetic-nZVI BC400 (BC400-m-nZVI), and magnetic-nZVI BC800 (BC800-m-nZVI).

| Materials | Yield (%) | pH | pH$_{ZPC}$ | Ash Content (%) | B.E.T. SSA (m²/g) |
|---|---|---|---|---|---|
| Bcgas | n.d. | 9.17 | 7.3 | 38.1 | 251.11 |
| BC400 | 42.1 | 9.37 | 6.95 | 30.3 | 141.23 |
| BC800 | 33.6 | 10.30 | 8.75 | 36.6 | 213.03 |
| nZVI | n.d. | n.d. | n.d. | n.d. | 158.27 |
| BCgas-m-nZVI | n.d. | n.d. | n.d. | n.d. | 192.07 |
| BC400-m-nZVI | n.d. | n.d. | n.d. | n.d. | 132.44 |
| BC800-m-nZVI | n.d. | n.d. | n.d. | n.d. | 181.92 |

n.d.—not determined; B.E.T. SSA—Brunauer–Emmett–Teller specific surface area.

The original biochars of BCgas, BC400, and BC800 have alkaline property with pH values varying from 9.17 to 10.30. These values are comparable with the pH value of 8.88 of a rice husk biochar produced at a pyrolytic temperature of 650 °C [15]. According to these authors, the pH value of biochar will increase with increasing pyrolytic temperature. This is because minerals begin to separate from the organic matrix when the ash content increases at temperatures above 350 °C [37]. The values of pH at the zero point charge ($pH_{ZPC}$) of the original biochars varied from 6.95 to 8.75. These are comparable with the reported $pH_{ZPC}$ of activated carbon of 7.75 [26]. The $pH_{PZC}$ corresponds to the pH value at which the surface of the solid is considered to be zero. It plays an important role during the sorption of ionic species on solid surfaces from aqueous systems [26]. In addition, $pH_{ZPC}$ may change due to the sorbent impregnation. Han et al. [36] reported that the $pH_{ZC}$ was decreased by the magnetite impregnation. Ash contents of the original biochars varied in the range from 30.3 to 38.1%. In general, high pyrolytic temperature will result in the volatilization of lignocellulose components [15,37] and this may result to the increase of ash content.

Morphology of the original materials and the modified materials: Morphology of the materials of nZVI, BC800, BC800-m, BC800-m-nZVI, BCgas, BCgas-m-nZVI, BC400, and BC400-m-nZVI was observed by the transmission electron micrographs (JOEL USA Inc., Peabody, MA, USA) at × 100,000 magnification (Figure 3). Figure 3a shows the morphology of nZVI particles which have flake-like shape with a longer side of approximately 20–50 nm and a narrower side of approximately 5–10 nm. This observation is different from the spherical shape of nZVI particles reported by Zhang, Jin [14]. Figure 3b shows the surface of a BC800 particle. Figure 3c shows how spherical magnetized iron oxide clusters were precipitated on BC800's surface. And Figure 3d shows the structure of BC800-m-nZVI where the magnetized iron oxide and nZVI clusters were both precipitated on BC800's surface. Similarly, Figure 3e–h show the structure of the materials of BCgas, BCgas-m-nZVI, BC400, and BC400-m-nZVI, respectively.

### 3.2. Color Removal Efficiency of the Modified Biochars for RY145, RR195, and RB19

The results show that the magnetic-nZVI biochars derived from both full-scale gasification (BCgas-m-nZVI) and lab-scale pyrolysis (BC400-m-nZVI and BC800-m-nZVI) significantly removed color of the dyeing stocks RY145, RR195 and RB19. The color removal efficiency of the modified biochars achieved the values of 100% for RY145 and RR195 and higher than 65% for RB19. Table 3 presents the color removal efficiency at a specific dose of the materials (g/L) where the efficiency reached a stable state. Figure 4 shows that the stable state of the modified biochars for RY145 was observed at doses of higher than 0.50 g/L. On the other hand, while the RY145 removal efficiency at this dose (0.50 g/L) of BCgas-m-nZVI was significantly higher ($p < 0.05$) than the one of BC400-m-nZVI (98.66 ± 0.15% *vs.* 93.89 ± 0.26%), it was not significantly higher than the one of BC800-m-nZVI (98.66 ± 0.15% *vs.* 98.44 ± 0.38%). Similarly, Figure 5 shows that the stable state of the modified biochars for RR195 was observed at doses of higher than 1.00 g/L. The RR195 removal efficiency at this dose (1.00 g/L) of BCgas-m-nZVI was significantly higher ($p < 0.05$) than the ones of BC400-m-nZVI and BC800-m-nZVI (94.14 ± 0.96% *vs.* 70.03 ± 1.67% and 86.31 ± 2.22%, respectively). Figure 6 shows that the stable state of the modified biochars for RB19 was observed at a dose of ≥ 6.00 g/L. The RB19 color removal efficiency at this dose (6.00 g/L) of BCgas-m-nZVI was significantly higher ($p < 0.05$) than the ones of BC400-m-nZVI and BC800-m-nZVI (76.84 ± 0.26% *vs.* 65.18 ± 0.27% and 69.72 ± 0.10%, respectively). Generally, color removal efficiency of the modified biochars derived from full-scale gasification method was higher than the one of the modified biochars derived from lab-scale pyrolysis method. This observation is also in agreement with the B.E.T. SSA of the modified biochars, achieving the values of 192.07, 132.44, and 181.92 $m^2$/g for BCgas-m-nZVI, BC400-m-nZVI, and BC800-m-nZVI, respectively (Table 2).

It is worth noting that, when doses of the magnetic-nZVI biochars were increased (≥ 0.50, 1.00, and 6.00 g/L for RY145, RR195, and RB19, respectively) then their color removal efficiency for RY145 and RR195 approached the highest value of 100% (Figures 4 and 5) or color removal efficiency for

RB19 continuously increased beyond 65% (Figure 6), while a similar trend was not observed in the case of nZVI material. For instance, when doses of nZVI exceeded the thresholds of 0.50 and 1.00 g/L then its color removal efficiency for RY145 and RR195 was lightly less (Figures 4 and 5). Particularly, color removal efficiency of nZVI for RB19 significantly decreased when its doses exceeded the value of 1.00 g/L. This is an interesting observation because, for nanoparticles like nZVI, when doses increase, they may aggregate together which reduces their reactivity (reducing accessible reactive surface area). The reason for impregnating nanoparticles on a biochar or similar carrier is often because it spatially separates the different nZVI particles and prevents their aggregation and therefore enhances their reactivity [10,13]. Furthermore, a decrease of color removal efficiency of nZVI material may be caused by the dissolution of the redundant nZVI amount into the dyeing solution when doses were too high. Similar phenomena were not observed in the treatments with the magnetic-nZVI biochars. These findings clearly illustrate the advantages of the magnetic-nZVI biochars.

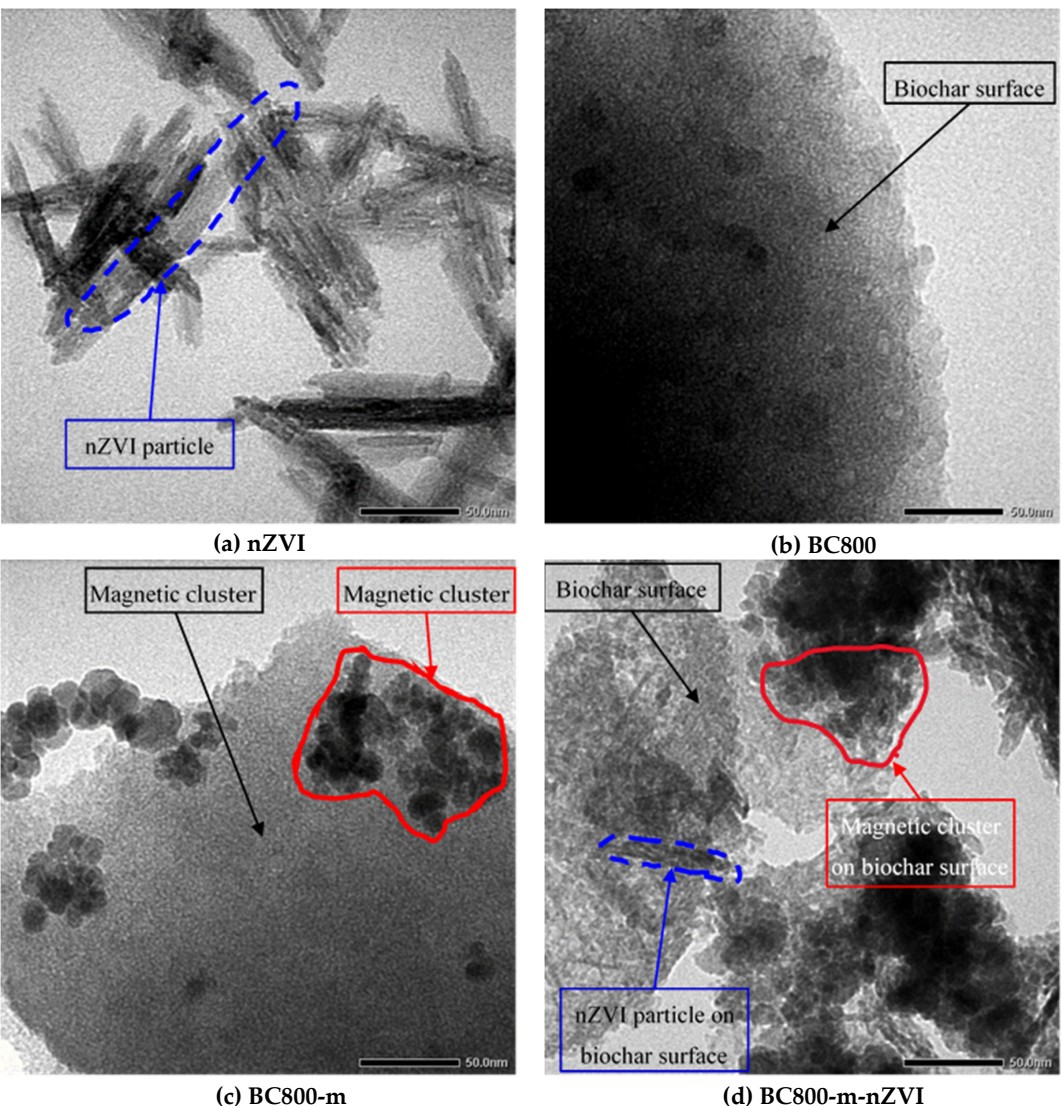

**Figure 3.** *Cont.*

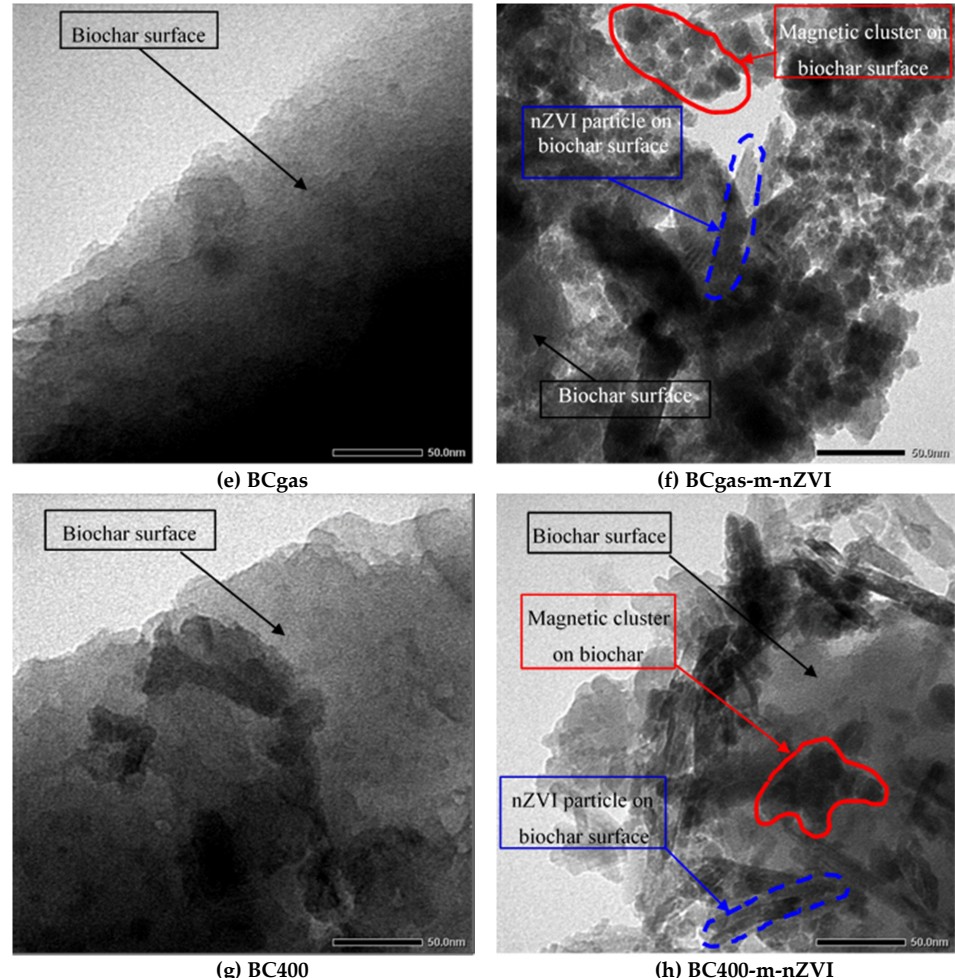

**Figure 3.** Transmission electron micrographs (JEM-1400 TEM, JOEL USA Inc., Peabody, MA, USA) at ×100,000 magnification (with the scale of 50.0 nm) of the materials of (**a**) nano zero valent iron (nZVI); (**b**) 800 °C—pyrolyzed rice husk biochar (BC800); (**c**) magnetic BC800 (BC800-m); (**d**) magnetic-nZVI BC800 (BC800-m-nZVI); (**e**) gasified rice husk biochar (BCgas); (**f**) magnetic-nZVI BCgas (BCgas-m-nZVI); (**g**) 400 °C—pyrolyzed rice husk biochar (BC400); and (**h**) magnetic-nZVI BC400 (BC400-m-nZVI).

**Table 3.** Color removal efficiency (η,%) of nano zero valent iron (nZVI), magnetic-nZVI gasified rice husk biochar (BCgas-m-nZVI), magnetic-nZVI 400 °C—pyrolyzed rice husk biochar (BC400-m-nZVI), and magnetic-nZVI 800 °C—pyrolyzed rice husk biochar (BC800-m-nZVI) for the reactive dyes yellow (RY145), red (RR195), and blue (RB19) at a specific dose of the materials (g/L).

| Materials | Color Removal Efficiency (η, %) at a Specific Dose of the Materials (g/L) | | |
|---|---|---|---|
| | **RY145** | **RR195** | **RB19** |
| nZVI | 94.62 ± 0.59 (at 0.25 g/L) | 77.66 ± 0.41 (at 0.50 g/L) | 21.40 ± 2.05 (at 1.00 g/L) |
| BCgas-m-nZVI | 98.66 ± 0.15 (at 0.50 g/L) | 94.14 ± 0.96 (at 1.00 g/L) | 76.84 ± 0.26 (at 6.00 g/L) |
| BC400-m-nZVI | 93.89 ± 0.26 (at 0.50 g/L) | 70.03 ± 1.67 (at 1.00 g/L) | 65.18 ± 0.27 (at 6.00 g/L) |
| BC800-m-nZVI | 98.44 ± 0.38 (at 0.50 g/L) | 86.31 ± 2.22 (at 1.00 g/L) | 69.72 ± 0.10 (at 6.00 g/L) |

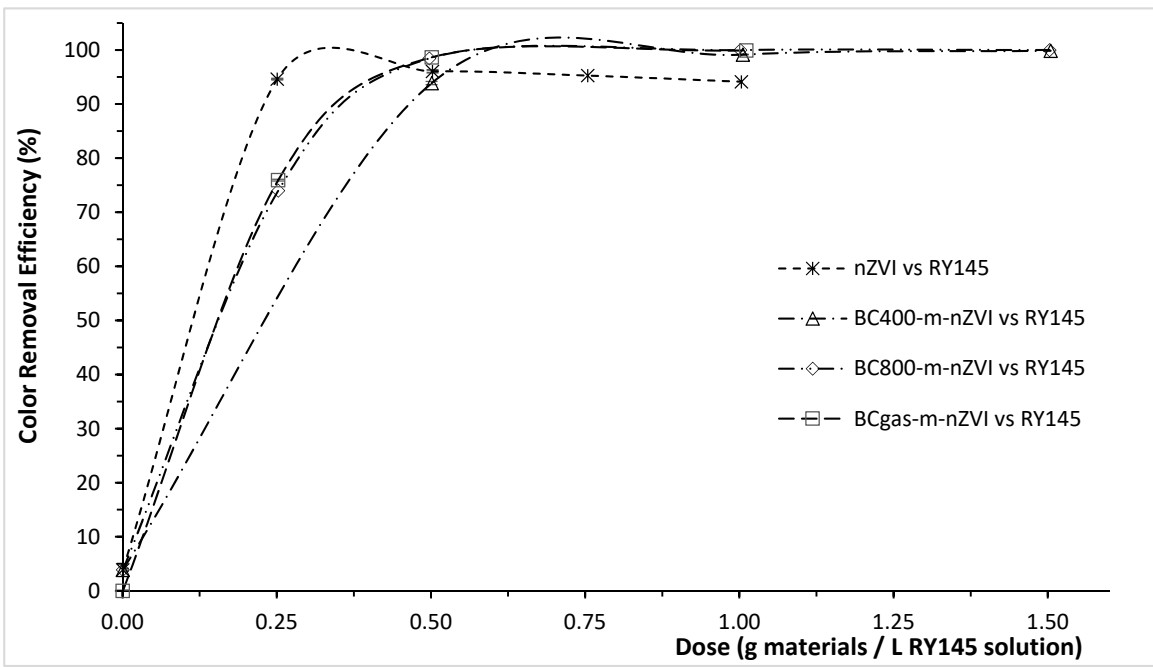

**Figure 4.** Color removal efficiency of the materials of nano zero valent iron (nZVI), magnetic-nZVI gasified rice husk biochar (BCgas-m-nZVI); magnetic-nZVI 400 °C—pyrolyzed rice husk biochar (BC400-m-nZVI); and magnetic-nZVI 800 °C—pyrolyzed rice husk biochar (BC800-m-nZVI) for reactive yellow dye (RY145) in dyeing solution.

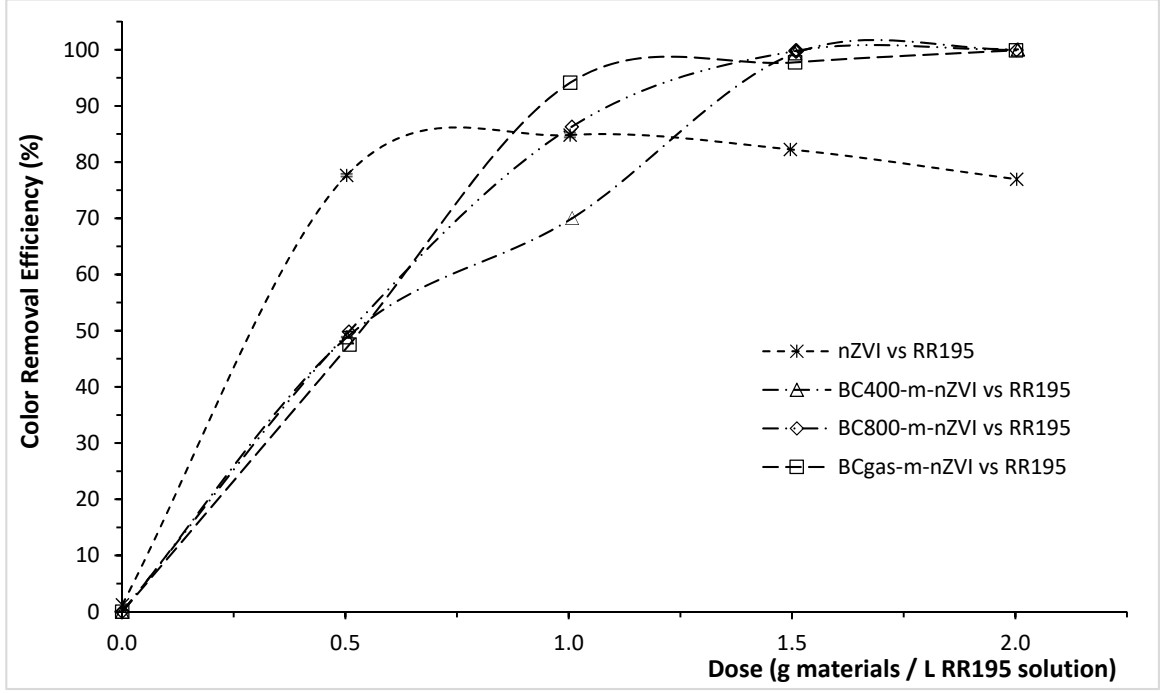

**Figure 5.** Color removal efficiency of the materials of nano zero valent iron (nZVI), magnetic-nZVI gasified rice husk biochar (BCgas-m-nZVI); magnetic-nZVI 400 °C—pyrolyzed rice husk biochar (BC400-m-nZVI); and magnetic-nZVI 800 °C—pyrolyzed rice husk biochar (BC800-m-nZVI) for reactive red dye (RR195) in dyeing solution.

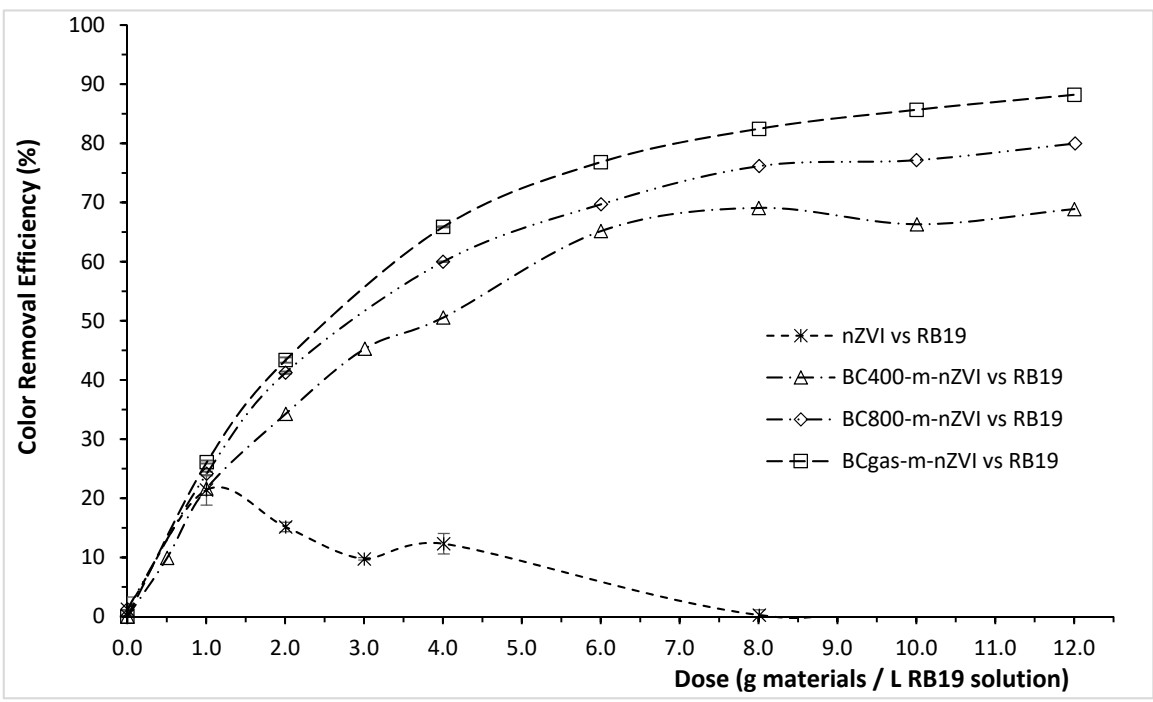

**Figure 6.** Color removal efficiency of the materials of nano zero valent iron (nZVI), magnetic-nZVI gasified rice husk biochar (BCgas-m-nZVI); magnetic-nZVI 400 °C—pyrolyzed rice husk biochar (BC400-m-nZVI); and magnetic-nZVI 800 °C—pyrolyzed rice husk biochar (BC800-m-nZVI) for reactive blue dye (RB19) in dyeing solution.

### 3.3. Color Removal Mechanisms of the Modified Biochars

Based on the results, Figure 7 (the continuous line with circle BC800 *vs*. RR195) presents that the original biochar BC800 can significantly absorb the dye RR195 with a color removal efficiency up to 57.16 ± 1.96% at a dose of 10.0 g/L. However, as a highly reactive chemical, nZVI plays a more important role than BC800 in reducing the color of the dyeing solutions [8,13]. Figure 7 presents that, at a low dose of 0.5 g/L, RR195 removal efficiencies of nZVI, BCgas-m-nZVI, BC400-m-nZVI, BC800-m-nZVI, and BC800 were determined to be 77.66 ± 0.41, 47.54 ± 0.57, 48.85 ± 4.3, 49.85 ± 0.76, and 4.99 ± 2.35%, respectively. These results show that, even though the original biochar BC800 can adsorb the dye RR195 with significantly lower efficiency, nZVI element was the main agent for reducing the color threshold of the dyeing solution. In this case, nZVI particles are good electron donors and dyeing molecules are excellent electron acceptors [38]. $Fe^0$ nanoparticles were reduced to $Fe^{2+}$ and $Fe^{3+}$ ions in the aqueous medium and the hydroxyl and/or hydrogen ions generated during reduction process react with dye molecules to induce the breaking of the chromophore $(-N = N-)$ bond [39,40]. The nZVI particles have to break the auxochrome bond as well to decolorize the dye molecules and the resulting intermediate organic compounds need further mineralization into $CO_2$, $H_2O$ and inorganic ions to achieve complete degradation [41]. This explains the higher efficiencies of the magnetic-nZVI biochars than their original material of biochar.

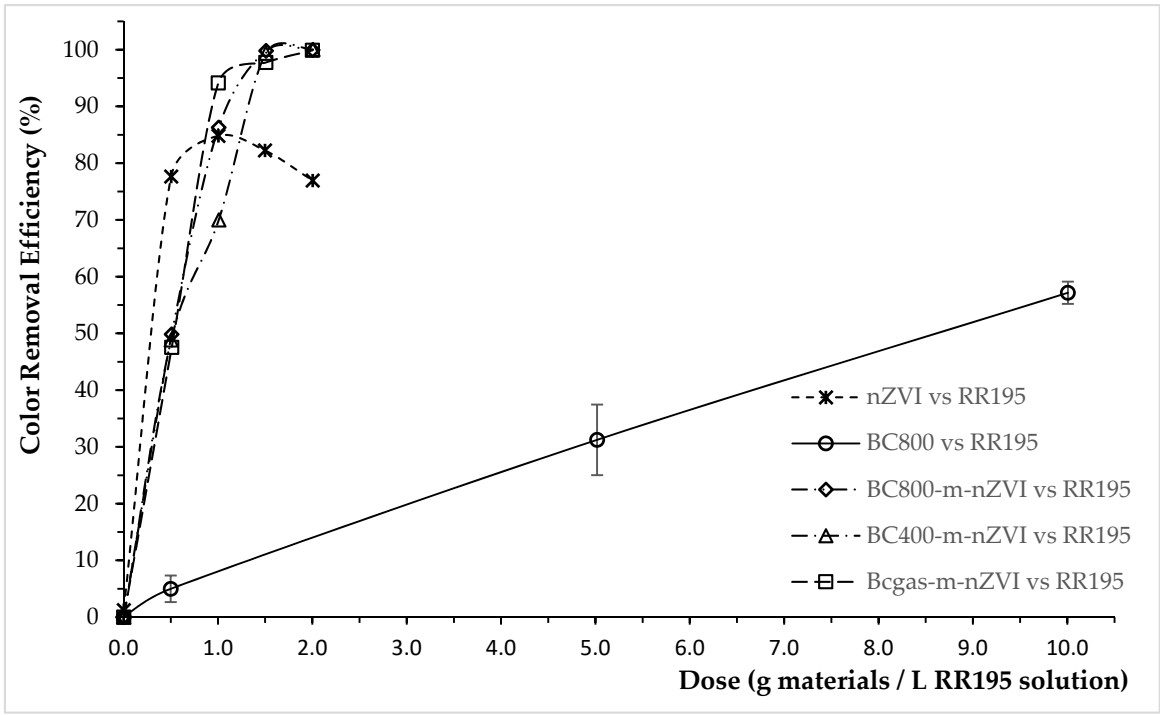

**Figure 7.** Color removal efficiency of the original materials of nano zero valent iron (nZVI) and 800 °C—pyrolyzed rice husk biochar (BC800), and the modified materials of magnetic-nZVI gasified rice husk biochar (BCgas-m-nZVI), magnetic-nZVI 400 °C—pyrolyzed rice husk biochar (BC400-m-nZVI), and magnetic-nZVI 800 °C—pyrolyzed rice husk biochar (BC800-m-nZVI) for reactive red dye (RR195) in dyeing solution.

As a natural polymers containing about 40% cellulose and 30% lignin [16], rice husk possess advantageous characteristics e.g., high molecular weight and high specific surface area for the decolorization process of dyeing wastewater treatment [42]. In this study, high specific surface area characteristics of biochars (Table 2) resulted in direct impregnation or sorption of nZVI particles on the biochar backbone surface. This prevents the aggregation process of nZVI particles and therefore enhances their reactivity.

## 4. Conclusions

The rice husk biochars modified with magnetized iron oxides and nZVI effectively reduced color of dyeing wastewater. Color removal efficiency of the modified biochars derived from both full-scale gasifier (BCgas-m-nZVI) and lab-scale pyrolytic furnace (BC400-m-nZVI and BC800-m-nZVI) achieved the values of 100% for dyes RY145 and RR195 and higher than 65% for dye RB19. When doses of the modified biochars were increased, their color removal efficiencies increased accordingly, while no such trend was observed in the case of the original material of nZVI. Color removal efficiency of the full-scale gasification modified biochars is higher than the one of the lab-scale pyrolytic modified biochars. The impregnation of nano particles of ZVI on the biochars' surface resulted in the spatial separation of different nZVI particles, prevented their aggregation and therefore improved the color removal efficiency. The results also showed the advantages of the magnetic-nZVI biochars over the unmodified biochars. While the original biochar material can adsorb dyes to some extent, the nZVI element impregnated on biochar was the main agent for reducing color of the dyeing solutions.

**Author Contributions:** Conceptualization, B.-S.T. and D.W.; methodology, B.-S.T. and D.W.; validation, B.-S.T. and P.T.K.L.; formal analysis, B.-S.T. and D.W.; investigation, B.-S.T., N.H.P.; writing—original draft preparation, B.-S.T.; writing—review and editing, B.-S.T., P.T.K.L., T.L.L. and D.W.; supervision, B.-S.T.; project administration, B.-S.T.; funding acquisition, B.-S.T.

**Funding:** This research was funded by VIETNAM NATIONAL UNIVERSITY HO CHI MINH CITY, grant number C2017-24-07.

**Acknowledgments:** Trinh acknowledges the Vietnam National University Ho Chi Minh city for funding this project. The authors also acknowledge the Ecofarm Co. LTd. in Long An, Vietnam for kindly supporting biomass (rice husk) and the full-scale gasification system.

**Conflicts of Interest:** The authors declare no conflict of interest. The funders had no role in the design of the study; in the collection, analyses, or interpretation of data; in the writing of the manuscript, or in the decision to publish the results.

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
