# Peer review of "Rice Husk Biochars Modified with Magnetized Iron Oxides and Nano Zero Valent Iron for Decolorization of Dyeing Wastewater"

_processes, doi:10.3390/pr7100660_

Round 1

Reviewer 1 Report

The manuscript written by Trinh et al in such shape and form is not suitable for publication in Processes or any other journal. In general, I miss serious literature review (39 references only) and discussion.

General remarks:

Abstract: Bigger picture not presented. Why is your research significant? What is so special about it? What was the motivation? Also, motivate the reader to continue reading your manuscript! Materials and methods: details missing for certain methods (e.g. microscopy). Supplemental materials can be moved to the main text.  Results: Not presented in a wider context (Discussion) or compared to other findings. Discussion: Missing. References: 39 references are not enough. Supplemental material: Everything can be moved to the main text. Figures: Microscopy: due to low resolution, I cannot se the scale. Please remind us in the caption about the scale.

Author Response

Point 1: The manuscript written by Trinh et al in such shape and form is not suitable for publication in Processes or any other journal. In general, I miss serious literature review (39 references only) and discussion. 

 Response 1: We appreciate and thanks for the valuable comments. The manuscript has been revised according to the reviewer’s comments. Three references have been added. Abstract, Introduction, Materials and Methods, Results and Discussion, and Conclusion sections have been rewritten and modified throughout the manuscript.

Point 2: Abstract: Bigger picture not presented. Why is your research significant? What is so special about it? What was the motivation? Also, motivate the reader to continue reading your manuscript!

Response 2: The abstract was revised to provide the bigger picture. Important findings were highlighted to grab the reader’s attraction.

Point 3: Materials and methods: details missing for certain methods (e.g. microscopy).

Response 3: Details of materials and methods have been added, modified and highlighted.

Point 4: Supplemental materials can be moved to the main text.

Response 4: Supplementary materials have been moved to the main text.

Point 5: Abstract: Bigger picture not presented. Why is your research significant? What is so special about it? What was the motivation? Also, motivate the reader to continue reading your manuscript!

Response 5: Results and Discussion have been modified in a wider context (highlighted text). Discussion was added in the same section of Results. Three important references and comparative findings have been added as per the reviewer’s request.

Point 6: Supplemental material: Everything can be moved to the main text. Figures: Microscopy: due to low resolution, I cannot se the scale. Please remind us in the caption about the scale.

Response 6: Supplemental materials were moved to the main text. The T.E.M micrographs have been replaced by higher resolution ones with a more clearly visible scale.

Reviewer 2 Report

Rice husk biochars modified with magnetized iron oxides and nano zero valent iron for decolorization of dying wastewater treatment.

Trinh et al.

Overall I believe that this is a nice piece of work and is publishable.  I have some specific questions and comments that I believe could be addressed by the authors.

Does the magnetization survive the complete preparation method?  I am surprised the authors did not demonstrate magnetic recovery of the particles.  This seems to be the point or preparing magnetic particles.  Otherwise why not just create the zero-valent iron on the chars directly as it appears that the magnetic mineral deposition decreased the surface area of the particles?

Was the presence of magnetic minerals in the final product confirmed by x-ray analysis? 

The authors report the pH of the resulting product and the pH of the zero point of charge for the staring material.  Does the zero point change after magnetization and creation of zero valent iron? 

It is not clear if pH was controlled during the de-colorization/adsorption experiments.  If the pH was lower than the zero point of charge it seems that this could promote the adsorption of the negatively charged dye molecules and perhaps increase de-colorization.  Was the impact of pH examined?

Why were different concentrations of the various dyes tested?  The RB19 was at a considerably higher concentration. This is a likely a partial explanation for lower de-colorization. 

Were all experiments conducted for only 1-hour?  Was this arbitrary?  Is there any time dependence for the de-colorization reaction to occur? 

The authors indicate that de-colorization was mainly accomplished by reduction of the azo-group on the dye molecules.  Can the contribution of adsorption to the process be quantified?  Was the formation of the reduction product quantified or quantifiable using UV data?  The authors utilized uv-vis to determine the fading of the color.  Presumably the spectra acquired would indicate if a product of the reduction remained in solution.  Otherwise, was total organic carbon measured in the reacted sample supernatant?  Was dissolved iron measurable in the supernatant after the reaction? 

Specific comments:

Line 62:  “the proportion of rice husk and bulk grain weight” do you mean the proportion of rice husk to bulk grain weight?

Line 154: sulfate spelling

Author Response

Point 1: Overall I believe that this is a nice piece of work and is publishable.  I have some specific questions and comments that I believe could be addressed by the authors. 

Response 1: Thanks very much for positive comments. Most of the questions and comments has been addressed and justified as follows below.

Point 2: Does the magnetization survive the complete preparation method?  I am surprised the authors did not demonstrate magnetic recovery of the particles.  This seems to be the point or preparing magnetic particles.

Response 2: The magnetization process was carefully conducted as an intermediate step of the biochar modification processes. The magnetic recovery was then tested by using a magnet. During the magnetization process, we used a magnet to separate and collect only the magnetized biochar particles.  The remaining particles which were not attached to the magnet were discarded along with the solution. Additional information (highlighted text in Section 2. Materials and Methods: Magnetic biochar:) has been added in the manuscript to make these points clearer.

Point 3: Otherwise why not just create the zero-valent iron on the chars directly as it appears that the magnetic mineral deposition decreased the surface area of the particles? 

Response 3: Previous studies have synthesized nZVI directly on the chars, such as (Tseng, Su et al. 2011, Liu and Wang 2019), while the doping of magnetic biochar with nZVI has not yet been demonstrated.

In addition, as stated in the introduction that “Application of magnetized carbon-based materials as adsorbents is of interest in industrial wastewater treatment processes because they can be easily separated with an external magnet from the treated water flow”.

It is likely correct that the magnetic mineral deposition may cause the decrease of the specific surface area.  Actually, the magnetic biochar has lower surface area, because magnetite has typically lower surface area than the biochar, so the composite has lower surface area than the pure biochar (Table 2 in the manuscript). However, Han, Sani et al. (2015) demonstrated that the carbonaceous surface area is largely preserved during the magnetite impregnation process.

Also, the results showed that nZVI (in both materials of nZVI only and the biochars modified nZVI) was the main agent which interacted and reacted with dyes for removing the color of the dyeing solution, while the original biochar (BC800) contributed a very low color removal efficiency (Figure 7 in the manuscript). The main biochar benefit is that it spaces out the nZVI, thus preventing its aggregation

Point 4: Was the presence of magnetic minerals in the final product confirmed by x-ray analysis?ʉ۬

Response 4: X-ray analysis is a reliable method for determining the presence of magnetic minerals in the final products. We acknowledge for the comment. However, due the real condition of our laboratory, the presence of magnetic minerals in the final products was confirmed by the use of a magnet.  Additional information has been added in the main text.

Point 5: The authors report the pH of the resulting product and the pH of the zero point of charge for the staring material. Does the zero point change after magnetization and creation of zero valent iron?ʉ۬

Response 5: We acknowledge that pHZPC may change due to the sorbent impregnation. Han, Sani et al. (2015) reported that the pHZC was decreased by the magnetite magnetite impregnation process. Additional information has been added in the main text.

Point 6: It is not clear if pH was controlled during the de-colorization/adsorption experiments.  If the pH was lower than the zero point of charge it seems that this could promote the adsorption of the negatively charged dye molecules and perhaps increase de-colorization.  Was the impact of pH examined?

Response 6:

Unfortunately, pH was not controlled during the decolorization experiments.  However, the influence of pH on the color removal efficiency of BC800-m-nZVI vs RR195 has been tested.  The results showed that, even with pH values as low as 2 – 4 resulted in high color removal efficiency, the high pH values of 5 – 7 was likely acceptable.

(please see the Figure of "Influence of pH on color removal efficiency of BC800-m-nZVI vs Reactive Red 195" in the attached file.

Point 7: Why were different concentrations of the various dyes tested?  The RB19 was at a considerably higher concentration. This is a likely a partial explanation for lower de-colorization. 

Response 7: It is because, as stated in the manuscript “The National Technical Regulation on Industrial Wastewater of Vietnam (QCVN-40:2011/BNTMT 2011) does not provide a specific concentration threshold (in mg/L) for a specific dye but it provides a specific color threshold (in Pt-Co) for industrial wastewater. For example, the maximum value of color parameter of industrial wastewater being discharged into receiving waters must be equal or lower than 50 Pt-Co for the receiver type A and 150 Pt-Co for the receiver type B. Therefore, the color removal efficiency of the materials for the reactive dyes was assessed by measuring the color in Pt-Co unit rather than absolute concentrations“.  Also, in reality, in the dyeing textile industry, various dyes with various concentrations could be used and discharged in the wastewater system.

Point 8: Were all experiments conducted for only 1-hour?  Was this arbitrary?  Is there any time dependence for the de-colorization reaction to occur? â€¨

Response 8: The reaction time was determined from the screening test. The period of 1 hour was chosen as an acceptable value. The dependence for the de-colorization reaction to occur is expressed in below figure.

(please see the Figure of "Influence of reaction time on color removal efficiency of BC800-m-nZVI vs Reactive Red 195" in the attached file.)

Point 9: The authors indicate that de-colorization was mainly accomplished by reduction of the azo-group on the dye molecules.  Can the contribution of adsorption to the process be quantified?  Was the formation of the reduction product quantified or quantifiable using UV data?  The authors utilized uv-vis to determine the fading of the color.  Presumably the spectra acquired would indicate if a product of the reduction remained in solution.  Otherwise, was total organic carbon measured in the reacted sample supernatant?  Was dissolved iron measurable in the supernatant after the reaction?  

Response 9: The hypothesis of decolorization of the dyes causing by the breaking the azo-group has been supported by literature (Shu, Chang et al. 2007, Fan, Guo et al. 2009).

Contribution of adsorption to the process can be quantified by using the same doses of BC800, nZVI and BC800-m-nZVI (Fig. 7 of the updated manuscript). For instances, as stated in Section 3.3, at a low dose of 0.5 g/L of BC800, nZVI and BC800-m-nZVI, RR195 removal efficiencies of these materials were determined to be 4.99 ± 2.35, 77.46 ± 0.41, and 49.85 ± 0.76 %, respectively. UV-VIS is a reliable method to confirm dye concentrations. There was a small amount of iron dissolved in the solution but it was below the Vietnamese discharge regulation for wastewater.

Point 10: Specific comments:

Line 62:  “the proportion of rice husk and bulk grain weight” do you mean the proportion of rice husk to bulk grain weight?

Line 154: sulfate spelling  

Response 10: Changed to “the proportion of rice husk to bulk grain weight”.

Corrected into “Sulfate”! With Thanks

Fan, J., Y. Guo, J. Wang and M. Fan (2009). "Rapid decolorization of azo dye methyl orange in aqueous solution by nanoscale zerovalent iron particles." Journal of Hazardous Materials 166(2-3): 904-910.

Han, Z., B. Sani, W. Mrozik, M. Obst, B. Beckingham, H. K. Karapanagioti and D. Werner (2015). "Magnetite impregnation effects on the sorbent properties of activated carbons and biochars." Water Research 70: 394-403.

Liu, X. and Y. Wang (2019). "Activated carbon supported nanoscale zero-valent iron composite: Aspects of surface structure and composition." Materials Chemistry and Physics 222: 369-376.

QCVN-40:2011/BNTMT (2011). National Technical Regulation on Industrial Wastewater of Vietnam. M. o. N. R. a. Environment.

Shu, H.-Y., M.-C. Chang, H.-H. Yu and W.-H. Chen (2007). "Reduction of an azo dye Acid Black 24 solution using synthesized nanoscale zerovalent iron particles." Journal of colloid and interface science 314(1): 89-97.

Tseng, H.-H., J.-G. Su and C. Liang (2011). "Synthesis of granular activated carbon/zero valent iron composites for simultaneous adsorption/dechlorination of trichloroethylene." Journal of Hazardous Materials 192(2): 500-506.

Reviewer 3 Report

The authors properly described in section 3.3 the removal with three different iron oxides. I feel that additional information should be added, at least discussed, concerning the potential ''direct'' interaction/adsorption between dye and rice/polysaccharides backbone. Clear comments should be stated on this. See

https://doi.org/10.1016/j.seppur.2019.115893

table 2

Author Response

Point 1: The authors properly described in section 3.3 the removal with three different iron oxides. I feel that additional information should be added, at least discussed, concerning the potential ''direct'' interaction/adsorption between dye and rice/polysaccharides backbone.

Clear comments should be stated on this. See https://doi.org/10.1016/j.seppur.2019.115893 table 2.

Response 1: The authors thank for valuable information and reference from the reviewer.

Several different iron oxides (Fe3O4), maghemite (Fe2O3, γ-Fe2O3) or zero valent iron (ZVI) might take part in the decolorization of dying solution. However, results (Fig. 7) showed that it was likely only ZVI that played an important role in reducing the color of the dyeing solution.

Additional information has been added in Section 3.3. as follows:

On the other hand, as a natural polymers containing about 40% cellulose and 30% lignin (Phonphuak and Chindaprasirt 2015), rice husk possesses advantageous characteristics e.g. high molecular weight and high specific surface area for the decolorization process of dyeing wastewater treatment (Lapointe and Barbeau 2020). In this study, high specific surface area characteristic of biochars (Table 2 of the manuscript) resulted in direct impregnation or sorption of nZVI particles on the biochar backbone surface. This prevents the aggregation process of nZVI particles and therefore enhances their reactivity.

Lapointe, M. and B. Barbeau (2020). "Understanding the roles and characterizing the intrinsic properties of synthetic vs. natural polymers to improve clarification through interparticle Bridging: A review." Separation and Purification Technology 231: 115893.

Phonphuak, N. and P. Chindaprasirt (2015). 6 - Types of waste, properties, and durability of pore-forming waste-based fired masonry bricks. Eco-Efficient Masonry Bricks and Blocks. F. Pacheco-Torgal, P. B. Lourenço, J. A. Labrincha, S. Kumar and P. Chindaprasirt. Oxford, Woodhead Publishing: 103-127.

Round 2

Reviewer 1 Report

The manuscript is in much better shape. However, I have two experimental concerns/questions. After the authors address these questions, I can endorse the manuscript for publication. 

If I understand the Materials and methods correctly, you prepared dye stocks with different concentrations. Why not the same?  Based on your methods, can you conclude that the dyes were really absorbed, and not chemically modified into colorless derivatives? I am not a chemist, but I think that desorption experiment or measuring the amount of remaining dissolved compounds after treatment could confirm/reject your "conclusion" there are some places with reduntant information (an example below) 

L27: "It is concluded" - please find an alternative.

L34-38: Too many things in parentheses. Please re-write the sentences.

L58: "could be considered" - is it not?

L120: tap or distilled water?

L126-7 and throughout: please use non-breaking space between values and units

L131: What were the dye concentrations?

L133: min, not mins

L134: what wavelength?

L134-142: What units? PCU? Are you talking about absorption maxima? or something else? Please rewrite and clarify the section about color. The method you are using is specific and limited to certain remediation fields, and some clarity for people who are not in your specific field would be appreciated.

L145: Please explain "DI" water the first mention it.

L154-157: This sentence does not belong to Materials and methods.

L170: above, in these references, or both? please specify

L190-192: Redundant. Mw described in table 1 AND in text

L194: typo - sulfate

L209-210: Spec described above.

Author Response

Round 2:

The manuscript is in much better shape. However, I have two experimental concerns/questions. After the authors address these questions, I can endorse the manuscript for publication. 

Point 1: If I understand the Materials and methods correctly, you prepared dye stocks with different concentrations. Why not the same? 

Response 1: We highly appreciate and thank the reviewer for this crucial question.  We considered this question before we designed and chose the methods for our experiments.

As stated in the revised main text at L211-214, the National Technical Regulation on Industrial Wastewater of Vietnam [1] does not provide a specific concentration threshold (in mg/L unit) for a specific dye but it provides a specific color threshold (in Pt-Co unit) for industrial wastewater.  For instances, the maximum values of color parameter of industrial wastewater being discharged into receiving waters in Colums A and B are 50 and 150 Pt/Co, respectivey.  According to this Regulation, the experiments were designed for investigation of the color removal efficiency of the modified biochar materials based on the color threshold (in Pt-Co unit) rather than the concentration threshold (in mg/L unit).

In our study, the dyeing stock solutions RY145, RR195, and RB19 were prepared with different concentrations of 11.2 10-6M, 24.5 10-6M, and 166.1 10-6M to achieve the same color thresholds of approximately 400Pt-Co.  Actually the colors of the stocks RY145, RR195, and RB19 were of 404.5, 408.8, and 410.9Pt-Co at  different wavelengths for maximum color absorbence of 419nm, 517nm, and 592nm, respectively (determined by an UV-VIS spectrometer analyzer - Shimazu UV1800). 

Based on your methods, can you conclude that the dyes were really absorbed, and not chemically modified into colorless derivatives? I am not a chemist, but I think that desorption experiment or measuring the amount of remaining dissolved compounds after treatment could confirm/reject your "conclusion" there are some places with reduntant information (an example below) 

Response 2: Figure 7 (the contineous line with circle BC800 vs RR195) in the results shows that the original, non-reactive biochar BC800 can significantly absorb the dye RR195 with a color removal efficiency up to 57.16 ± 1.96% at a dose of 10.0g/L. However, as stated in L372-376 "as a high reactive chemical, nZVI plays a more important role than BC800 in reducing the color of the dyeing solution [2, 3].  Figure 7 presents that, at a low dose of 0.5g/L, RR195 removal efficiencies of nZVI, BCgas-m-nZVI, BC400-m-nZVI, BC800-m-nZVI, and BC800 were determined to be 77.66 ± 0.41, 47.54 ± 0.57, 48.85 ± 4.3, 49.85 ± 0.76, and 4.99 ± 2.35%, respectively".

We therefore report that both the original biochar (BC800) and the modified biochar (BC800-m-nZVI) can significantly reduce the color of the dyeing solutions. However, the reactive component nZVI played a more important role than BC800 in the color reduction.

L27: "It is concluded" - please find an alternative.

Response 3: It was replaced by “It is reported that …” (yellow highlighted).

L34-38: Too many things in parentheses. Please re-write the sentences.

Response 4: The parentheses have been removed and replaced by the commas.

L58: "could be considered" - is it not?

Response 5: The term has been replaced by “Rice husk possesses …” (yellow highlighted).

L120: tap or distilled water?

Response 6: The term “deionized water” has been added (yellow highlighted).

L126-7 and throughout: please use non-breaking space between values and units

Response 7: This comment has been addressed throughout the main text.

L131: What were the dye concentrations?

Response 8: The initial color concentrations of the stocks have been added (yellow highlighted).

L133: min, not mins

Response 9: This comment has been addressed

L134: what wavelength?

Response 10: The wavelengths have been added (yellow highlighted).

L134-142: What units? PCU? Are you talking about absorption maxima? or something else? Please rewrite and clarify the section about color. The method you are using is specific and limited to certain remediation fields, and some clarity for people who are not in your specific field would be appreciated.

Response 11: The unit of the dyeing stocks is Pt-Co. The text of this section has been rewritten (yellow highlighted).

L145: Please explain "DI" water the first mention it.

Response 12: the term “DI water” has been replaced by “deionized water”

L154-157: This sentence does not belong to Materials and methods.

Response 13: This sentence has been rewritten for matching to the Materials and Methods section (yellow highlighted).

L170: above, in these references, or both? please specify

Response 14: The sentence has been corrected and the references have been removed because they were listed in the “nano zero valent iron” section (yellow highlighted).

L190-192: Redundant. Mw described in table 1 AND in text

Response 15: The information of molecular weights has been removed as it was presented in Table 1 (yellow highlighted).

L194: typo - sulfate

Response 16: The term “sunfate” has been corrected to “sulfate”

L209-210: Spec described above.

Response 17: This comments has been addressed.

QCVN-40:2011/BNTMT, National Technical Regulation on Industrial Wastewater of Vietnam, M.o.N.R.a. Environment, Editor. 2011. Raman, C.D. and S. Kanmani, Textile dye degradation using nano zero valent iron: A review. Journal of Environmental Management, 2016. 177: p. 341-355. Tseng, H.-H., J.-G. Su, and C. Liang, Synthesis of granular activated carbon/zero valent iron composites for simultaneous adsorption/dechlorination of trichloroethylene. Journal of Hazardous Materials, 2011. 192(2): p. 500-506.

Round 3

Reviewer 1 Report

The authors addressed all of my questions and concerns. I can endorse the manuscript for publication after the authors correct minor remarks listed below.

throughout the manuscript: please make sure that there is non-breaking space (https://en.wikipedia.org/wiki/Non-breaking_space) between every value and every unit. Right now, there are NO spaces.

throughout the manuscript: please make sure that you use en dashes, em dashes, and hyphens correctly (in combination with spaces). Please proofread your manuscript, and correct where needed.

An example of wrong use: L38

Current style: number, space, hyphen, space, number, no space, unit

Correct: number, en dash, number, non-breaking space, unit.

Author Response

Round 3:

The authors addressed all of my questions and concerns. I can endorse the manuscript for publication after the authors correct minor remarks listed below.

Point 1: throughout the manuscript: please make sure that there is non-breaking space (https://en.wikipedia.org/wiki/Non-breaking_space) between every value and every unit. Right now, there are NO spaces. 

Response 1: The non-breaking space between the value and the unit has been applied.

Point 2: throughout the manuscript: please make sure that you use en dashes, em dashes, and hyphens correctly (in combination with spaces). Please proofread your manuscript, and correct where needed. 

An example of wrong use: L38

Current style: number, space, hyphen, space, number, no space, unit

Correct: number, en dash, number, non-breaking space, unit.

Response 2:    Hyphen, en dash, em dash rule has been applied and corrected throughout the manuscript. It was also proofreaded and modified for better understanding. 
